# Applicability and Freshness Control of pH-Sensitive Intelligent Label in Cool Chain Transportation of Vegetables

**DOI:** 10.3390/foods12183489

**Published:** 2023-09-19

**Authors:** Tianlin Feng, Huizhi Chen, Min Zhang

**Affiliations:** 1State Key Laboratory of Food Science and Resources, Jiangnan University, Wuxi 214122, China; 17762004211@163.com (T.F.); chenhuizhiht@163.com (H.C.); 2China General Chamber of Commerce Key Laboratory on Fresh Food Processing & Preservation, Jiangnan University, Wuxi 214122, China; 3Jiangsu Province International Joint Laboratory on Fresh Food Smart Processing and Quality Monitoring, Jiangnan University, Wuxi 214122, China

**Keywords:** intelligent label, color change, anthocyanin, pH indicator, freshness detection model

## Abstract

Freshness is one of the main factors affecting consumers’ purchase of food. The freshness indicator labels of packaged fresh green bell pepper (*Capsicum annuum* L.) and greengrocery (*Brassica chinensis* L.) were constructed, and pH-sensitive indicator labels based on the dye of anthocyanin and the mixing dye of methyl red and bromothymol blue were prepared in this study. At the same time, the color, chlorophyll content and vitamin C content of vegetables were measured in order to explore the applicability of indicator labels in the cool chain transportation of vegetables. Compared with the nature dye, the chemical dye-type indicator labels are more sensitive to pH changes. The results showed that the mixed indicator intelligent label had the best indication effect, and the MB 2 (mixing 1 g/L methyl red and bromothymol blue solutions at a ratio of 3:2 with a concentration of 70 mL/L in indicator film solution) indicator label could effectively indicate the freshness changes in vegetables during storage. Meanwhile, the color changes of the MB 2-type indicator label were correlated with the colors change of the sample, changes in nutrients, and changes in CO_2_ content inside the packaging. In addition, freshness detection models for green bell pepper and greengrocery by using color information of MB 2 intelligent labels were established. Hence, this pH-sensitive label can be applied as a promising intelligent packaging for non-destructively monitoring the freshness of respiratory and non-respiratory climacteric vegetables.

## 1. Introduction

The freshness of fruit and vegetables has significantly affected consumers’ purchase intention, so it is very important to maintain high quality and visualize the quality changes of fruit and vegetables in the fresh food supply chain [1,2]. In addition, changes in the sensory quality of stored food also affect consumers’ purchasing desire [3]. Intelligent packaging is defined as using the communication function of the packaging system to communicate the conditions of packaged food [4,5]. A typical freshness package can show the freshness of the goods inside the package and inadequate microbiological quality through the color change directly observed by the naked eye [6]. This kind of indicator label is generally based on the change of microorganism level of the foods in the package, different storage conditions or physiological phenomena during storage [7].

There has been an endless stream of research on the preservation of fresh products, and researchers have gradually turned their research toward the intelligent detection of product quality [8,9,10,11]. The indicator system in intelligent packaging is usually attached or added to the packaging material [12]. For example, Suzuki et al. used polyethylene active film to detect the change of activity of fresh-cut apples [13]. Since postharvest fruit and vegetables will still perform respiration, the pH in the package will change, so the use of pH sensors and indicators can effectively indicate the change of freshness [14]. Among them, pH-sensitive indicators are popular due to their convenient preparation and intuitive display. The typical freshness indicator label based on pH value mainly consists of two parts: solid carrier and pH-sensitive indicator material [15]. Studies have found that color-based pH-sensitive indicators are a promising intelligent packaging, and they have been used in the freshness detection of meat [16], fish [17], shrimp [18] and other products. Among them, colorimetric dyes of chemical indicator type are widely used and can provide consumers with food information more accurately [19,20]). In recent years, people have become more and more interested in natural dyes. Because these natural dyes are non-toxic and harmless, more environmentally friendly [21]. At present, the commonly used substrates mainly include biopolymers such as starch, polyvinyl alcohol, cellulose and carrageenan [22,23]. In the past research, there have been many research studies on intelligent packaging for meat, aquatic product, and fresh-cut fruit and vegetables. Meanwhile, there are few research studies on intelligent packaging for fresh fruit and vegetables, especially non-respiratory climacteric fruit and vegetables. Some scholars have studied the freshness indicator labels of fresh green bell peppers [19].

In this study, available smart labels were prepared for the freshness study of respiratory climacteric vegetables (such as fresh green peppers) and non-respiratory climacteric plants (such as greengrocery). Mainly based on the respiration of vegetables during storage, which leads to changes in pH value, the color of indicator label also changes accordingly, thus playing an indicative role.

## 2. Materials and Methods

### 2.1. Preparation of Vegetables

The green bell pepper and greengrocery used in the experiment were purchased from a local supermarket. After the vegetables were washed and dried, they were packed in sealed polyethylene terephthalate (PET) boxes. At least 12 boxes were needed in the experiment, so we prepared 30 boxes to ensure the smooth progress of the experiment. The PET box was a buckle type, and after the overall lid was closed, it was in a sealed state. The sample weight in each box was about 300 g. The inner surface of the PET box cover was pasted with intelligent labels which had not contacted the packaged vegetables. After packaging, samples were stored at 4 °C, and the relevant indexes were measured at 0, 3, 6, 9, 12 and 15 days after storage. The packaged vegetables were shown in Figure 1.

### 2.2. Preparation of pH-Sensitive Intelligent Label

#### 2.2.1. Chemical Dye-Type Indicator Label

First, 1 g/L of methyl red solution in solvent of alcohol solution (100%) (recorded as MR) and 1 g/L bromothymol blue solution in solvent of alcohol solution (100%) (recorded as B) were prepared and uniformly mixing MR and B in the volume ratio of 3:2 (recorded as MB). Then, 20 g/L glycerol (plasticizer) and 30 g/L methylcellulose (adhesive) were added into the solution. Solution was taken in a beaker and put in a 70 °C water bath until the mixing solution was stirred evenly. Then, a certain amount of indicator solution was added and stirred at high speed to form a uniform gel solution, and the pH was adjusted to about 7.5 [19]. The indicator solution and pH are shown in Table 1.

After ultrasonic defoaming (JY98-3D ultrasonic cleaning machine: Zhejiang Ningbo Xinzhi Biotechnology Co., Ltd., Ningbo, China), the gel solution was poured into the petri dish, dried at 45 °C for 8 h in a hot airdrying oven, and prepared into the intelligent label indicator film. The indicator film was cut into 1 × 1 cm size and tightly wrapped with food-grade polyethylene film as the background to obtain the indication label. The label can be used after being pasted on the inner surface of the PET packaging box cover.

#### 2.2.2. Anthocyanin Indicator Labels

We dissolved 4 g of starch and 2 g of polyvinyl alcohol (PVA) in 20 mL of distilled water at 95 °C and then diluted it to 100 mL with distilled water, respectively, and then mixed them with a 1:1 volume ratio. Then, 1% mulberry extract and 30% glycerol on the basis of the weight of the mixed solution were added into the solution. The obtained solution was defoamed by ultrasonic treatment (JY98-3D ultrasonic cleaning machine: Zhejiang Ningbo Xinzhi Biotechnology Co., Ltd., Ningbo, China), poured on the petri dish, dried at 35 °C for 48 h, and the intelligent label indicator film containing anthocyanins was obtained. We named this indicator film D-H. The intelligent label was prepared by the same method as “Chemical dye-type indicator label”.

We put 1 g of chitosan into 100 mL 1% acetic acid and stirred it in a magnetic stirrer to obtain a chitosan solution with a concentration of 1%. The solution was adjusted to neutral (pH 6~7) by 1 mol/L NaOH solution. Then, 0.1 g of mulberry anthocyanin (obtained by extraction in the laboratory using the extraction method in powder form) was dissolved in the above solution and stirred evenly to obtain chitosan/anthocyanin solution. Afterwards, 70 mL of chitosan/anthocyanin solution was transferred to a square dish and dried at 35 °C for 48 h. Then, we removed it and named K-H. The intelligent label was prepared by the same method as the last paragraph of “Chemical dye-type indicator label”.

#### 2.2.3. Usage of Intelligent Label

In the experiment, different kinds of intelligent labels were prepared by the method explained above, and the use of intelligent labels was monitored within 15 days. When placing the sample, there was no contact between the sample and the label, and there was a gap of about 1 cm in the middle. The indication label was placed in the middle of the PET box cover and inside the packaging box (Figure 1).

### 2.3. Anthocyanin Content of Mulberry Extract

First, 0.1 g of mulberry extract was taken, extracted with 1% hydrochloric acid solution until colorless, and then filtered to 100 mL. The extract was properly diluted; 5 mL of the diluted extract was taken and added into a 25 mL colorimetric tube, and 10 mL of 0.025 mol/L KCl-HCl buffer (pH = 1.0) was added into the 25 mL colorimetric tube. Another 5 mL of extract was added into a 25 mL colorimetric tube, and 10 mL of 0.4 mol/L NaAc-HCl buffer solution (pH = 4.5) was added, too. Distilled water was taken as the control. The absorbance value was measured at 520 nm and 700 nm. The calculation formula of anthocyanin content was as follows:(1)Anthocyanin content(mgg)=A×MW×DF×1000ε
where *DF* is the dilution ratio, *MW* is 484.82 g/mol (the relative molecular weight of cornflower 3-glucoside), ε = 24,825 L·mol^−1^·cm^−1^ (the molar extinction coefficient of cornflower 3-glucoside at 510 nm), and *A* is absorbance. The results are expressed in mg/L cornflower 3-glucoside equivalent.

### 2.4. pH-Dependent Discoloration Ability

The label was immersed in a buffer solution with a pH of 3–11 for 3 min. After the buffer solution was removed, the label was put in a place with uniform illumination to take photos and record the color change [24].

### 2.5. CO_2_ Concentration Detection Inside the Packaging

A gas detector (Shanghai Ruifen International Trade Co., Ltd., Shanghai, China) was used to measure the CO_2_ concentration inside the packaging, and the results were expressed in the form of percentage (*v*/*v*).

### 2.6. Quality Test of Vegetable Samples

#### 2.6.1. Color

The color of samples was measured with a hand-held colorimeter (Konica Minolta, CR-400, Tokyo, Japan), which was calibrated according to the standard calibration plate on the white surface. There was a circular mark on the skin of each group of five samples to ensure that color differences are tested at the same location. Each sample was measured 5 times, and the average value was taken as the final result. These five measurements randomly selected different parts and fixed these five positions for subsequent measurements. Using the CIE color system, *L**, *a** and *b** values of the sample were automatically obtained by a colorimeter and used as the indexes to evaluate the color quality of samples.

#### 2.6.2. Vitamin C (VC) Content

The content of VC in the sample was determined by potassium iodate titration. First, 10 g of thawed sample was put into mortar; then, 2% (*v/v*) hydrochloric acid solution was added and ground into slurry in an ice bath. The slurry was transferred into a 100 mL volumetric flask, and the volume was adjusted to the scale with 2% hydrochloric acid solution. After 10 min of extraction, the suspension was filtered and the filtrate was reserved.

Then, 0.5 mL of 10 g/L potassium iodide solution, 2.0 mL of 5 g/L starch solution, 5.0 mL of extract and 2.5 mL of distilled water were added into a conical flask and mixed well. We titrated with 1 mmol/L potassium iodate until the solution was blue and did not fade within 30 s. The volume of potassium iodate solution was recorded three times. Then, 5.0 mL of 2% hydrochloric acid solution was taken as a blank control and titrated according to the same method [25]. The content of VC was calculated as follows:(2)$$\mathrm{VC}\mathrm{content}(\mathrm{mg}/100\;g)=\frac{V\times \left(V1-V0\right)\times 0.088}{Vs\times m}\times 100$$
where *V*1 is the volume of potassium iodate solution consumed in sample titration (mL), *V*0 is the volume of potassium iodate solution consumed in blank titration (mL), *Vs* is the volume of sample solution taken during titration (mL), *V* is the total volume of the sample extract (mL), *m* is the mass of the sample (g), and 0.088 is the mass of *VC* equivalent to 1 mL l mmol/L potassium iodate solution.

#### 2.6.3. Chlorophyll Content

The representative sample was cut and mixed, and then, it was pounded with the tissue crusher into homogenate for standby. Then, 0.5 g of sample was weighed into a triangular flask, and we added 100 mL of extraction agent (anhydrous ethanol:acetone = 1:1). The triangular flask was sealed at room temperature, kept away from light for 5 h, and then filtered. The extraction agent was used as the blank and zero, and the absorbance was measured at 663 nm (*A*1) and 645 nm (*A*2), respectively [26]. The calculation formula of chlorophyll content was as follows:(3)Chlorophyll content(mgg)=(8.05×A1+20.29×A2)×V1000×m

### 2.7. Sensory Evaluation

Fifteen volunteers (7 women and 8 men, aged 23–28) participated in the sensory test of this experiment. All 15 volunteers have received at least one sensory test training before understanding the scoring criteria (Table 2). The sensory characteristics (taste, flavor, texture and appearance) and overall acceptability of the samples were evaluated by 5 points [27].

### 2.8. Color Change of Indicator Label

The color change of the indicator label in the process of use was measured by a hand-held colorimeter (Konica Minolta, CR-400, Tokyo, Japan), and the result was expressed by the △*E* value. During the test, the indicator label was wrapped in a plastic film for measurement, which would not affect the subsequent use.
(4)$$△E=\sqrt[{2}]{{\left(L^{*}\right)}^{2}+{\left(a^{*}\right)}^{2}+{\left(b^{*}\right)}^{2}}$$

### 2.9. Principal Component Analysis (PCA)

PCA was performed on the color information (R, G, B, *L**, *a**, *b**) of the MB 2 indicator label. We extracted the chromaticity data of indicator labels from photos using MATLAB R2014a to separate RGB (red, green, and blue values) and CIE Lab (*L**, *a**, *b** values) coordinates.

The position of each sample in the principal component can be intuitively seen through the principal component analysis graph, and the samples can be classified and processed. When the cumulative contribution rate of the principal components exceeds 85%, it can better represent the information of the original data. When the cumulative total contribution rate reaches 70%~85%, principal components can be used to reflect the overall information [28].

### 2.10. Statistical Analysis

Statistix 9.0 was used for one-way ANOVA, and a Tukey multiple-comparison test was used for significance analysis. The different letters in the figure and table indicated significant differences between samples (*p* < 0.05), and the data were represented by mean ± standard deviation. GraphPad Prism 8.0 was used for drawing. Origin 2021 was used for principal component analysis. SPSS 16.0 was used for Fisher discriminant analysis. Number of training (modeling) samples: The number of predicted samples was 2:1. When modeling, we chose to leave one method as the cross-validation method.

Fisher discriminant analysis was conducted on the color data of MB 2 indicator labels at different storage times to investigate the matching of different color gradients with the freshness of green bell pepper and greengrocery. There were 18 samples in the modeling group and 9 samples in the prediction group.

## 3. Results and Discussion

### 3.1. pH-Dependent Color-Changing Ability

Figure 2 shows the color change of different kinds of intelligent labels in pH 3–11. It is worth noting that different intelligent labels had different pH-dependent discoloration capabilities, which is due to the different composition and content of substances in these labels. Other studies have also shown similar results with different component labels having varying pH-dependent discoloration abilities [29]. The content of anthocyanins in the mulberry extract reached 173.45 ± 0.84 mg/g. The pH-dependent color change of anthocyanins is due to their structural changes in buffer solutions with different pH values, which is well proved by the research results of other researchers [24,30].

Figure 3 shows the changes in the use of intelligent labels after packaging. Among them, the labels of MB 1 and MB 2 presented the most obvious color changes, gradually changing from orange to red. However, the two labels containing anthocyanins did not show obvious color changes. Therefore, from the perspective of actual use, MB 1 and MB 2 labels performed best.

### 3.2. CO_2_ Concentration Detection Inside the Packaging

The changes in CO_2_ concentration inside the packaging of green bell pepper and greengrocery are shown in Figure 4. Due to the fact that green bell peppers were stored under refrigeration conditions in this study, refrigeration temperatures reduced their respiratory rate and delayed the emergence of their respiratory peaks. The respiration of green bell pepper after harvest showed a gradual upward trend, and the gas exchange inside and outside the packaging was blocked, resulting in the CO_2_ concentration in the packaging gradually increased compared with the beginning. This was also related to the color change of the label; as the CO_2_ concentration inside the packaging increased, the color of the label gradually deepened. Greengrocery is a kind of non-respiratory climacteric vegetable. After picking, the respiratory intensity of greengrocery gradually weakened, resulting in a slow increase in the concentration of CO_2_ inside the packaging. However, the concentration of CO_2_ reached 0.5% at the end of storage. Due to the insignificant change in CO_2_ concentration inside the packaging of greengrocery, which was less than 0.5%, the color change of greengrocery’s intelligent label during storage was not as significant as that of green bell pepper, with only the MB 2-type intelligent label showing a significant color change.

### 3.3. Quality of Vegetable Samples

#### 3.3.1. VC Content and Chlorophyll Content

In terms of the nutrition of these two vegetables, vitamin C and chlorophyll were selected. During the storage, as the freshness of vegetables decreased, the nutrients in them were also lost. It can be seen from Figure 5 and Figure 6 that the nutrient content of both of them gradually decreased during storage. At the end of storage, the chlorophyll content of green bell pepper decreased. This can also be seen from the color changes of its leaves. The changes in nutrients corresponded to color changes. This is also consistent with previous research findings [31]. Nutrients are gradually lost as the quality of vegetables deteriorates, so changes in vegetable nutrients can also be expressed through the color representation of intelligent tags.

#### 3.3.2. Sensory Evaluation

It can be seen from Table 3 that the sensory quality of green bell pepper and greengrocery decreased to different degrees during storage, and the degree of decline of greengrocery was greater. In general, when the score is lower than three, it is difficult to arouse people’s desire to buy. In terms of sensory characteristics, green bell pepper deteriorated slowly in the early stage, and only after long-term storage did it decline significantly. The overall acceptability of green bell pepper decreased during storage, and there was a more significant decline after 12 days of storage; however, greengrocery had a continuous downward trend in the storage process, and the overall performance was unacceptable at the end of storage. For green bell pepper, the color change of intelligent label was not very obvious in the early stage, but it was obvious in the later stage. For greengrocery, the changes were slow during storage. The results of other researchers indicate that for meat such as pork [29], the change in indicator labels is more pronounced, while for vegetables, especially non climacteric vegetables, the indicator effect is not very significant [32].

### 3.4. Color Change of Vegetable Samples and Indicator Label

Some green bell pepper had some visible quality changes, including petiole decay and green bell pepper skin shrinkage. In this study, at the end of the 15-day label observation, it can be found that the quality of greengrocery has deteriorated significantly, including the yellowing of leaves and corruption of some leaves. The specific changes are shown in Table 4. It showed that the quality of the two vegetables changed significantly during the storage at 4 °C.

The pH-sensitive indicator label we prepared will change color with the change of pH in this process. From Table 5, it can be seen that the color of the label changes with the freshness of the vegetables, and the trend of the two changes is correlated. Among them, the most significant change was the MB 2 indicator label, which was followed by the MB 1 indicator label. The D-H and K-H indicator labels also had measurable color changes, but these changes were not obvious to the naked eye. The results showed that the MB 2 indicator label was the best intelligent label.

### 3.5. Qualitative Discrimination of Freshness Level of Green Bell Pepper and Greengrocery Based on pH Sensitive Intelligent Packaging Label (MB 2) Color

#### 3.5.1. Principal Component Analysis (PCA)

PCA was performed on the color information (R, G, B, *L**, *a**, *b**) of the MB 2 indicator label, and the results are shown in Figure 7. Based on the freshness evaluation analysis above, the principal component analysis results of green bell pepper showed that the sample data on the 0th and 3rd days were a group, corresponding to fresh samples; the sample data on the 6th and 9th days were a group, corresponding to the second fresh sample; the sample data on the 12th and 15th days were a group, corresponding to spoiled samples. The total contribution rate of Principal Component 1 and Principal Component 2 was 83.9%. The PCA results indicated that the MB 2 indicator label can better distinguish the freshness of green bell pepper during storage.

Based on the freshness evaluation analysis above, the principal component analysis results of greengrocery showed that the sample data on the 0th and 3rd days were a group, corresponding to fresh samples; the sample data on the 6th day was divided into a separate group, corresponding to the second fresh sample; the sample data on the 9th, 12th, and 15th days were a group, corresponding to spoiled samples. The total contribution rate of Principal Component 1 and Principal Component 2 was 94.1%. The PCA results indicated that the MB 2 indicator label can better distinguish the freshness of greengrocery during storage.

#### 3.5.2. Fisher Discriminant Analysis

The Wilk’s Lambda values of the six variables were all small, indicating that the differences between the groups of each variable were statistically significant (*p* < 0.01), and the group mean was unequal, indicating that discriminant analysis was meaningful. The Fisher discriminant function coefficients (non-standardized coefficients) of green bell pepper are shown in Table 6. From Figure 7, it can be seen that the contribution rate of discriminant function 1 was 95.7%, and the contribution rate of discriminant function 2 was 4.3%. This result was consistent with the judgment of the freshness level of green bell pepper based on quality indicators. The established discriminative model of green bell pepper freshness grade was as follows:
*Y*(Fresh) = 0.312 × R − 0.597 × G − 0.501 × B − 0.143 × *L** + 2.829 × *a** + 1.298 × *b** − 102.127(5)
*Y*(Second fresh) = 0.256 × R − 0.436 × G − 0.611 × B−0.054 × *L** + 2.678 × *a** + 1.388 × *b** − 112.821(6)
*Y*(Spoiled) = 0.452 × R − 0.252 × G − 0.571 × B − 0.200 × *L** + 2.098 × *a** + 1.209 × *b** − 137.854(7)

The results of using the above discriminant function to classify the freshness of green bell pepper are shown in Table 7. The freshness of the samples on the 0th and 3rd day of refrigeration was judged as fresh with a recognition rate of 100% for the modeling set and 100% for the prediction set. The freshness of the samples on the 6th and 9th day of refrigeration was judged as second freshness, with a recognition rate of 83.33% for the modeling set and 100% for the prediction set. The freshness of the samples on the 12th and 15th days of refrigeration was judged as stable, with recognition rates of 100% for both the modeling and prediction sets. Therefore, the freshness of green bell pepper during storage can be divided into three levels: fresh, second fresh, and spoiled. The recognition rate of the modeling set was 100%, and the recognition rate of the prediction set was 88.89%.

The Wilk’s Lambda values of the six variables were all small, indicating that the differences between the groups of each variable were statistically significant (*p* < 0.01), and the group mean was unequal, indicating that discriminant analysis was meaningful. The Fisher discriminant function coefficients (non-standardized coefficients) of greengrocery are shown in Table 6. From Figure 7, it can be seen that the contribution rate of discriminant function 1 was 96.9%, and the contribution rate of discriminant function 2 was 3.1%. This result was consistent with the judgment of the freshness level of greengrocery based on quality indicators. The established discriminative model for the freshness grade of greengrocery was as follows:*Y*_(Fresh)_ = 4.184 × R + 1.673 × G + 0.124 × B + 0.311 × *L** − 0.418 × *a** − 0.699 × *b** − 827.405(8)
*Y*(Second fresh) = 1.796 × R + 0.864 × G + 0.451 × B + 0.170 × *L** − 0.461 × *a** − 0.081 × *b** − 220.092(9)
*Y*(Spoiled) = 0.524 × R + 1.209 × G + 1.007 × B + 0.336 × *L** + 0.436 × *a** + 0.607 × *b** − 278.377(10)

The results of using the above discriminant function to classify the freshness of greengrocery are shown in Table 7. The freshness of the samples on the 0th and 3rd day of refrigeration was judged as fresh with a recognition rate of 100% for the modeling set and 100% for the prediction set. The freshness of the samples on the 6th day of refrigeration was judged as second freshness with a recognition rate of 100% for the modeling set and 100% for the prediction set. The freshness level of the samples on the 9th, 12th, and 15th day of refrigeration was determined as stable, and the recognition rates of the modeling and prediction sets were both 100.0%. Therefore, through the color information and discriminant analysis of the MB 2 indicator label, the freshness of greengrocery during storage can be divided into three levels: fresh, second fresh, and spoiled. The recognition rate of the modeling set was 100%, and the recognition rate of the prediction set was 100%.

## 4. Conclusions

In this study, a variety of intelligent labels for detecting the quality changes of fresh vegetables have been developed. The results showed that the intelligent label made of the mixture of MR and BB at a 3:2 ratio (MB 2 formula) as the indicator and glycerol and methylcellulose as the matrix was most suitable for the freshness minoring of cool chain transportation of green bell pepper and greengrocery. The relationship between the color changes of MB 2 and the storage quality (color and nutrients) showed similar trends. In addition, since the label containing anthocyanins prepared in this experiment was not ideal, further work was needed to find suitable natural dyes to improve the safety of the label. In addition, the application of intelligent labels on other fruit and vegetables needs further research, especially non-climacteric fruit and vegetables.

## Figures and Tables

**Figure 1 foods-12-03489-f001:**
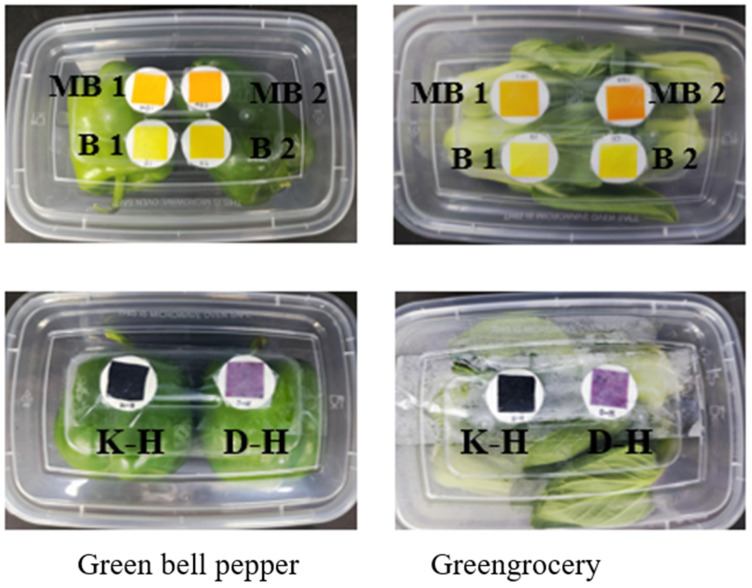
Intelligent label packaging display. (B 1: 50 mL/L B; B 2: 70 mL/L B; MB 1: 50 mL/L MB; MB 2: 70 mL/L MB; K-H: chitosan and anthocyanin indicator label; D-H: starch and anthocyanin indicator label).

**Figure 2 foods-12-03489-f002:**
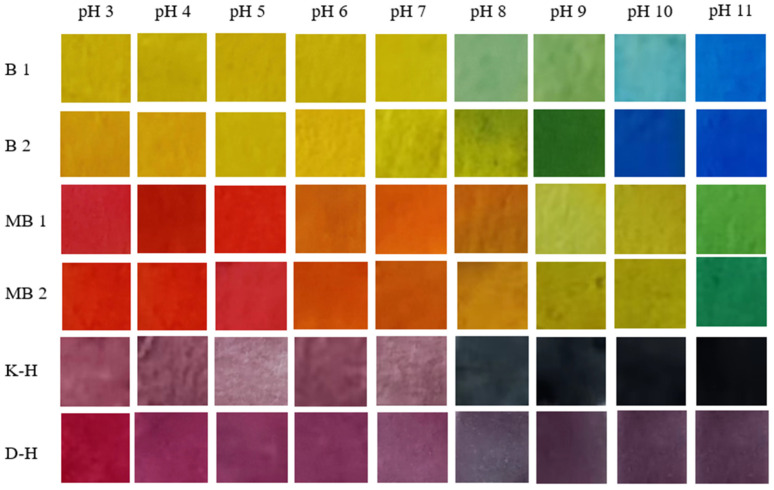
The pH-dependent color-changing ability of intelligent labels (B 1: 50 mL/L B; B 2: 70 mL/L B; MB 1: 50 mL/L MB; MB 2: 70 mL/L MB; K-H: chitosan and anthocyanin indicator label; D-H: starch and anthocyanin indicator label).

**Figure 3 foods-12-03489-f003:**
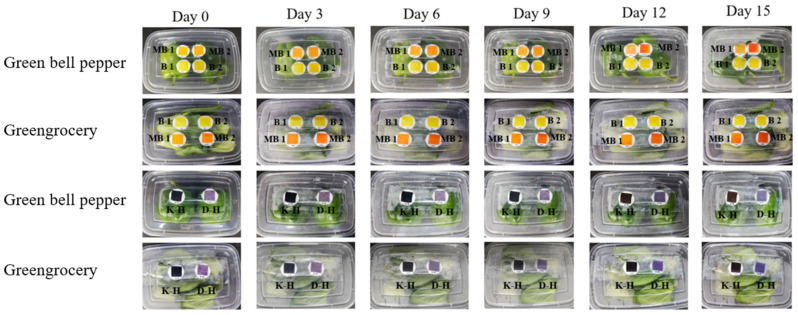
Color change of intelligent label during storage (B 1: 50 mL/L B; B 2: 70 mL/L B; MB 1: 50 mL/L MB; MB 2: 70 mL/L MB; K-H: chitosan and anthocyanin indicator label; D-H: starch and anthocyanin indicator label).

**Figure 4 foods-12-03489-f004:**
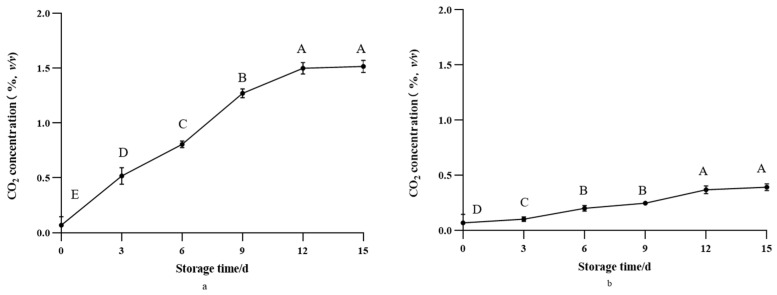
Changes in CO_2_ concentration inside packaging of green bell pepper and greengrocery during storage, (**a**) green bell pepper, (**b**) greengrocery. The bars represent standard deviations. Different letters in the figure indicate significant differences between samples (*p* < 0.05).

**Figure 5 foods-12-03489-f005:**
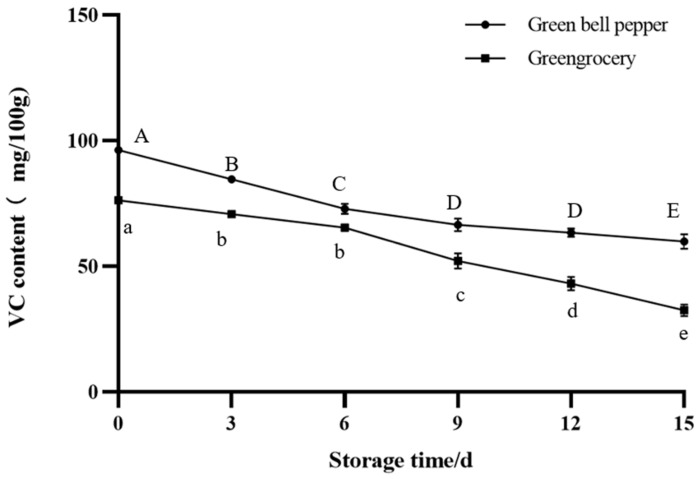
VC content of vegetables during storage. The bars represent standard deviations. Different letters in the figure indicate significant differences between samples (*p* < 0.05).

**Figure 6 foods-12-03489-f006:**
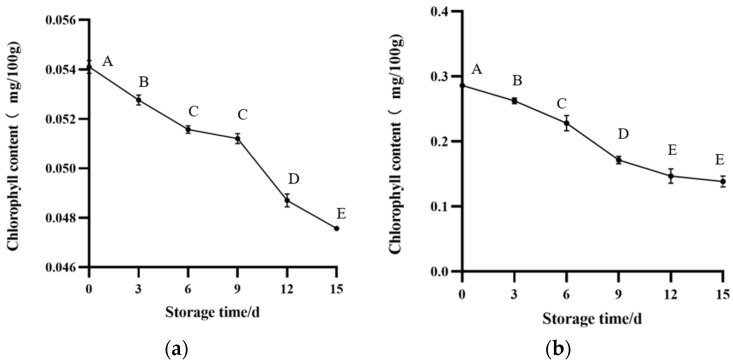
Chlorophyll content of vegetables during storage: (**a**) green bell pepper, (**b**) greengrocery. The bars represent standard deviations. Different letters in the figure indicate significant differences between samples (*p* < 0.05).

**Figure 7 foods-12-03489-f007:**
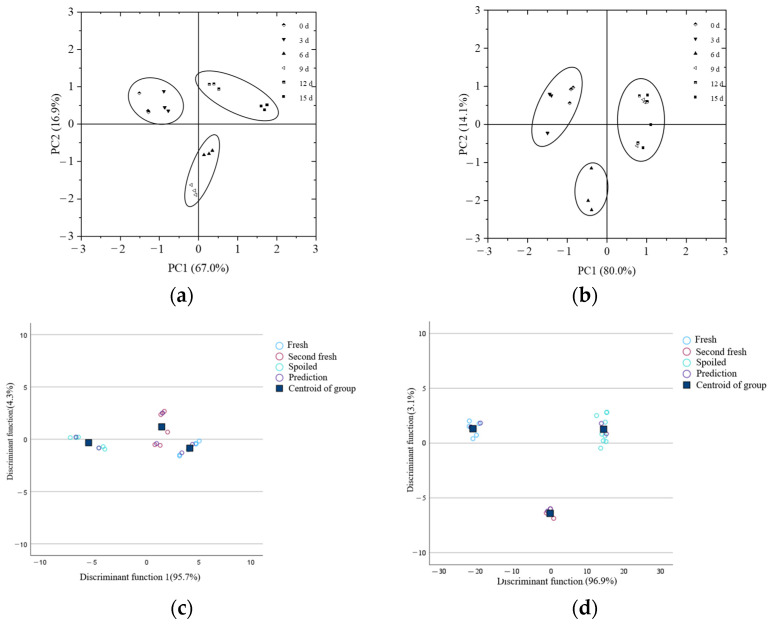
Principal component analysis and Fisher discriminant analysis diagram of color indicators for green bell pepper and greengrocery during storage (**a**–**c**): green bell pepper, (**b**–**d**): greengrocery.

**Table 1 foods-12-03489-t001:** Addition amount of label indicator and required pH of solution.

Intelligent Label	Concentration	Initial pH of Matrix
B 1	B 50 mL/L	7.50 ± 0.05
B 2	B 70 mL/L	7.50 ± 0.05
MB 1	MB 50 mL/L	7.50 ± 0.05
MB 2	MB 70 mL/L	7.50 ± 0.05

**Table 2 foods-12-03489-t002:** The standard score sheet for the sensory evaluation.

Score	Taste	Flavor	Texture	Appearance	Overall Acceptability
5	Refreshing, juicy and sweet; appropriate brittleness	Special aroma; favorable soft and comfortable	Complete fruit tissue; stiff and springy	Full flesh; no drip loss	Excellent
4	Less sweet or juicy; a certain degree of brittleness	Special aroma; relatively soft and comfortable	Certain springy	Full flesh; a little drip loss	Good
3	Lighter sweetness; general brittleness	Special aroma	Slightly soft	Partly wrinkled; a little drip loss	General
2	No sweetness; tender	A little special aroma	Soft	Partly wrinkled; serious drip loss	Bad
1	No sweetness; soft rotten	Pungent odor	Rotten	Sever wrinkled; serious drip loss	Unacceptable

**Table 3 foods-12-03489-t003:** Variations in the sensory scores of green bell pepper and greengrocery during storage.

		Day 0	Day 3	Day 6	Day 9	Day 12	Day 15
Taste	Green bell pepper	5	4.8	4.5	3.9	3.5	2.8
	Greengrocery	5	4.7	4.3	3.3	2.3	1.3
Flavor	Green bell pepper	5	4.8	4.4	4.0	3.6	2.5
	Greengrocery	5	4.7	4.3	3.5	2.1	1.1
Texture	Green bell pepper	5	4.8	4.5	3.8	3.4	2.5
	Greengrocery	5	4.8	4.2	3.3	2.0	1.2
Appearance	Green bell pepper	5	4.9	4.5	4.0	3.4	2.8
	Greengrocery	5	4.8	4.3	3.4	2.4	1.0
Overall Acceptability	Green bell pepper	5	4.8	4.6	4.0	3.4	2.8
	Greengrocery	5	4.7	4.3	3.4	2.2	1.1

**Table 4 foods-12-03489-t004:** Color of green bell pepper and greengrocery during storage.

		Green Bell Pepper	Greengrocery
*L**	Day 0	40.56 ± 0.53 ^B^	39.62 ± 1.32 ^C^
	Day 3	42.21 ± 1.01 ^AB^	40.13 ± 0.76 ^AB^
	Day 6	43.13 ± 0.76 ^A^	45.37 ± 1.27 ^B^
	Day 9	41.88 ± 0.33 ^B^	45.58 ± 0.33 ^B^
	Day 12	42.51 ± 1.12 ^AB^	63.22 ± 1.12 ^A^
	Day 15	44.25 ± 0.87 ^A^	65.54 ± 0.83 ^A^
*a**	Day 0	−15.72 ± 0.71 ^A^	−10.64 ± 0.31 ^B^
	Day 3	−15.34 ± 1.01 ^A^	−10.91 ± 0.78 ^B^
	Day 6	−13.22 ± 0.53 ^B^	−11.25 ± 1.03 ^B^
	Day 9	−12.97 ± 0.33 ^B^	−13.21 ± 0.31 ^AB^
	Day 12	−12.35 ± 0.72 ^B^	−15.75 ± 0.53 ^A^
	Day 15	−12.11 ± 0.43 ^BC^	−16.88 ± 0.23 ^A^
*b**	Day 0	6.78 ± 0.51 ^C^	11.71 ± 0.35 ^C^
	Day 3	7.13 ± 0.63 ^C^	12.79 ± 0.58 ^C^
	Day 6	8.24 ± 0.33 ^C^	13.25 ± 0.73 ^C^
	Day 9	10.17 ± 0.31 ^BC^	20.54 ± 0.41 ^B^
	Day 12	14.98 ± 0.52 ^B^	28.07 ± 0.17 ^A^
	Day 15	18.08 ± 0.23 ^A^	30.13 ± 0.53 ^A^

The data are presented as mean ± standard deviation (*n* = 5). Different letters in the same column indicate significant differences between samples (*p* < 0.05).

**Table 5 foods-12-03489-t005:** Total color difference (ΔE) of indicator labels for green bell pepper and greengrocery.

∆E							
		Day 0	Day 3	Day 6	Day 9	Day 12	Day 15
B 1	Green bell pepper	5.26 ± 0.23 ^F^	6.74 ± 1.65 ^E^	7.83 ± 0.57 ^D^	8.54 ± 1.42 ^C^	9.13 ± 0.48 ^B^	10.25 ± 0.82 ^A^
	Greengrocery	5.53 ± 1.01 ^E^	6.04 ± 0.78 ^D^	7.36 ± 0.48 ^C^	7.65 ± 0.78 ^B^	8.52 ± 0.52 ^A^	8.64 ± 0.43 ^A^
B 2	Green bell pepper	5.13 ± 1.21 ^E^	5.69 ± 1.43 ^D^	6.53 ± 0.76 ^C^	6.74 ± 1.14 ^C^	7.12 ± 0.74 ^B^	8.23 ± 1.01 ^A^
	Greengrocery	5.25 ± 0.93 ^D^	5.53 ± 0.63 ^D^	6.17 ± 1.43 ^C^	6.23 ± 0.89 ^C^	6.54 ± 0.77 ^B^	7.05 ± 0.47 ^A^
MB 1	Green bell pepper	6.47 ± 0.83 ^F^	7.53 ± 0.87 ^E^	8.75 ± 1.18 ^D^	15.33 ± 0.76 ^C^	23.46 ± 0.43 ^B^	30.57 ± 1.33 ^A^
	Greengrocery	6.32 ± 0.76 ^E^	6.59 ± 0.42 ^E^	7.23 ± 1.24 ^D^	7.94 ± 1.04 ^C^	10.57 ± 0.58 ^B^	20.73 ± 1.21 ^A^
MB 2	Green bell pepper	6.73 ± 1.32 ^F^	8.52 ± 1.35 ^E^	12.33 ± 0.92 ^D^	20.94 ± 1.13 ^C^	28.65 ± 0.37 ^B^	40.57 ± 1.14 ^A^
	Greengrocery	6.57 ± 0.78 ^F^	7.03 ± 1.21 ^E^	8.94 ± 1.41 ^D^	14.37 ± 0.83 ^C^	20.13 ± 0.56 ^B^	26.12 ± 1.09 ^A^
K-H	Green bell pepper	10.89 ± 0.43 ^C^	10.37 ± 0.82 ^C^	11.26 ± 0.87 ^B^	11.54 ± 0.47 ^B^	11.29 ± 1.01 ^B^	12.01 ± 0.23 ^A^
	Greengrocery	11.06 ± 1.13 ^B^	10.87 ± 1.04 ^C^	10.96 ± 0.42 ^B^	11.25 ± 0.79 ^B^	11.33 ± 1.13 ^A^	11.16 ± 0.21 ^A^
D-H	Green bell pepper	8.78 ± 0.48 ^D^	8.83 ± 0.33 ^D^	8.78 ± 1.11 ^D^	9.03 ± 0.51 ^C^	9.12 ± 0.43 ^B^	10.16 ± 0.25 ^A^
	Greengrocery	8.13 ± 0.92 ^D^	8.05 ± 0.51 ^D^	8.23 ± 0.28 ^C^	8.26 ± 1.23 ^C^	9.13 ± 0.71 ^A^	9.29 ± 0.42 ^A^

The data are presented as mean ± standard deviation (*n* = 5). Different letters in the same row indicate significant differences between samples (*p* < 0.05).

**Table 6 foods-12-03489-t006:** Fisher discriminant function coefficients based on color coordinates of MB 2 indicator labels for freshness discrimination of green bell pepper and greengrocery.

	Variable	Discriminant Function 1	Discriminant Function 2
Green bell pepper	R	−0.036	−0.003
	G	0.077	0.027
	B	0.013	0.060
	*L**	0.001	0.003
	*a**	−0.034	0.034
	*b**	0.005	−0.048
	Constant	2.725	−5.146
Greengrocery	R	0.000	0.021
	G	0.024	0.071
	B	0.037	0.020
	*L**	−0.102	0.029
	*a**	−0.013	0.072
	*b**	0.025	0.025
	Constant	11.808	−19.248

**Table 7 foods-12-03489-t007:** Classification of the freshness of green bell pepper and greengrocery using Fisher discriminant function.

Sample		Modeling Set					Prediction Set				
		Fresh	Second Fresh	Stable	Total	Recognition Rate (%)	Fresh	Second Fresh	Stable	Total	Recognition Rate (%)
Green bell pepper	Refrigeration Day 0 and Day 3	6	0	0	6	100	3	0	0	3	100
	Refrigeration Day 6 and Day 9	0	6	0	6	100	1	2	0	3	66.67
	Refrigeration Day 12 and Day 15	0	0	6	6	100	0	0	3	3	100
	Total				18	100				9	88.89
Greengrocery	Refrigeration Day 0 and Day 3	6	0	0	6	100	3	0	0	3	100
	Refrigeration Day 6	0	3	0	3	100	0	3	0	3	100
	Refrigeration Day 9, Day 12 and Day 15	0	0	9	9	100	0	0	3	3	100
	Total				18					9	100

## Data Availability

Data is contained within the article.

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
