# Peer review of "Applicability and Freshness Control of pH-Sensitive Intelligent Label in Cool Chain Transportation of Vegetables"

_foods, 2023, doi:10.3390/foods12183489_

Round 1

Reviewer 1 Report

The aim of the study was to investigate the effects of applicability and freshness control of pH-sensitive intelligent label in cool chain transportation of vegetables.

The manuscript has two major shortcomings. The first is that the statistics section is very poorly written. Secondly, there is little to no discussion. The manuscript should definitely be enriched by discussion.

The plural of fruit is fruit. The word fruits should be corrected as fruit.

L 91: How many boxes were prepared?

L 175: Color measurement in fruit or vegetables is usually measured in different directions. Color measurement of vegetables was done 5 times. These measurements were made 5 times on different sides of the vegetables. If done, it should be stated in the text.

L 176: Chroma and Hue angle from color measurements made should be calculated with equations. These color parameters are important for consumer preference.

L 217: The statistical analysis part of the study is very weak. Authors should provide more detailed explanations for the reader. In the study, the first Normal distribution and homogeneous tests are applied to the data for statistical analysis. According to the result obtained from this, parametric or non-parametric analysis is performed. However, these were not mentioned in this study. According to which criteria statistical analyzes were made. Authors should explain in detail.

L 294: It should be stated under the heading of statistical analysis in the material and method section of the PCA analysis.

L 294: PCA 1 and PCA 2 should be used as abbreviations in the text.

L 313: No explanation is given in the statistics section on Fisher discriminant analysis.

Figs. In 3, 4 and 5, statistical lettering should be given by making multiple comparisons.

Again, under all Tables and Figures, it should be given that “±” is standard error and standard deviation.

If there are significant differences between the parameters examined, it should be explained in the text that they are according to P<0.05 or P<0.01.

Apart from the specified (minor) things, the quality of the English language is entirely satisfactory.

Author Response

The aim of the study was to investigate the effects of applicability and freshness control of pH-sensitive intelligent label in cool chain transportation of vegetables.

  1. The manuscript has two major shortcomings. The first is that the statistics section is very poorly written. Secondly, there is little to no discussion. The manuscript should definitely be enriched by discussion.

Response: Thank you for your suggestion. We have provided additional details of statistics section and made improvements of discussion section in the article.

  1. The plural of fruit is fruit. The word fruits should be corrected as fruit.

Response: Thank you for your suggestion. We have corrected the mistake in the article.

  1. L 91: How many boxes were prepared?

Response: Thank you for your question. We have provided additional explanations in the corresponding positions of the article. Please see lines 91 in the revised manuscript.

  1. L 175: Color measurement in fruit or vegetables is usually measured in different directions. Color measurement of vegetables was done 5 times. These measurements were made 5 times on different sides of the vegetables. If done, it should be stated in the text.

Response: Thank you very much for your suggestion. We have provided additional details of color measurement. Please see lines 175 in the revised manuscript.

  1. L 176: Chroma and Hue angle from color measurements made should be calculated with equations. These color parameters are important for consumer preference.

Response: Thank you very much for your suggestion. The instrument used adopted the CIE color system, which we have explained in the article.

  1. L 217: The statistical analysis part of the study is very weak. Authors should provide more detailed explanations for the reader. In the study, the first Normal distribution and homogeneous tests are applied to the data for statistical analysis. According to the result obtained from this, parametric or non-parametric analysis is performed. However, these were not mentioned in this study. According to which criteria statistical analyzes were made. Authors should explain in detail.

Response: Thank you very much for your suggestion. We have made explanations in the corresponding positions of the article.

  1. L 294: It should be stated under the heading of statistical analysis in the material and method section of the PCA analysis.

Response: Thank you very much for your suggestion. We have made explanations in section 2.9 of the article.

  1. L 294: PCA 1 and PCA 2 should be used as abbreviations in the text.

Response: Thank you very much for your suggestion. We have made correction in the article.

  1. L 313: No explanation is given in the statistics section on Fisher discriminant analysis.

Response: Thank you very much for your suggestion.

  1. Figs. In 3, 4 and 5, statistical lettering should be given by making multiple comparisons.

Response: Thank you very much for your suggestion.

  1. Again, under all Tables and Figures, it should be given that “±” is standard error and standard deviation.

Response: Thank you for your suggestion. We have made additions and improvements in the article.

  1. If there are significant differences between the parameters examined, it should be explained in the text that they are according to P<0.05 or P<0.01.

Response: Thank you for your suggestion. We have made additions and improvements in the article.

Reviewer 2 Report

This manuscript gives interesting idea in developing intelligent packaging with natural dyes but proves they don't work well with vegetables. The findings are interesting but the text in general need major revision before publishing. Here is the list of more specific remarks:

Lines 21 and 22 please rephrase to be more understandable

Lines 27 MB you have to introduce the whole name before using abbreviations

Line 45-48 please rephrase, the sentence is too long.

Line 48-51 is the repetition of the previous sentence

Line 51-53 improve the English

Line 57 please change direction to with towards

Line 59 please delete to after attached

Line 60-61 should be rephrased, For example Suzuki et al used polyethylene...

Line 62 pH sensitive indicator labels are pH sensors, right?

Line 76-80 please rephrase in more English manner

Line 81-83 the goal or aim of the research should be stressed and point out novelty, not just a single statement.

Line 91-93 please modify and explain better.

Line 96-102 should be deleted from here because this is explanation and not experimental procedure

Line 103-105 please rewrite, it is ethanolic or alcohol solution of MR was prepared by dissolving 1g of MR in 1 L of how much was the ethanol percentage and so on...

Line 106-109 also explain better, you can not say certain amount you should give the exact amount, then it was heated to 70 degrees or just the bath was at 70 degrees, then how long you kept it and so on.

Line 112 culture dish do you mean Petri dish?

Line 113 intelligent label indicator film can you use one uniform name? there is no need to put extra words to sound fancy.

Line 118 anthocyanidin should be with small letter

Line 118-119 it is not procedure, please delete

Line 120 mulberry extract, please explain did you buy this extract or you prepared it yourself? If you did prepare it explain how, is it in liquid or powder state?

Line 121-123 please explain how did you prepare the PVA and starch solutions.

Line 123 please delete rich in anthocyanin and be consistent whether you will use glycerin or glycerol

Line 124 is the total weight of the mixed solution or the biopolymers weight because you are supposed to add plasticizers in respect to the polymer and not the weight/volume of the solution.

Line 124 please change defoaming with defoamed

Line 125 please specify is the glass mold a Petri dish or other type of mold and is the glass non-sticky and can you peel of the film easily?

Line 127-128 when you mention that it was prepared like previous method, is this about the part of 1cm2 squares that you cut out? explain better

Line 129 please give the exact amount and write 1% acetic acid

Line 134 what do you mean by baked? did you mean dry/ dried?

Line 140 please change shown with explained or written

Line 140-142 please delete this sentence, it is result or statement and not procedure

Line 147 it should be the extract was properly diluted...

Line 162 how did you record the color change? did you use colorimeter, what type?

Line 165 %(v/v) of what?

Line 167-169 please delete, not procedure

Results in 3.2 ;3.3 and 3.4 need comparison with other research

Line 366-368 please rewrite

Line 375-381 please rewrite and please dont start consecutive sentencec in the same way.

A native speaker of English or someone who speaks well English should revise the text.

Author Response

This manuscript gives interesting idea in developing intelligent packaging with natural dyes but proves they don't work well with vegetables. The findings are interesting but the text in general need major revision before publishing. Here is the list of more specific remarks: 1. Lines 21 and 22 please rephrase to be more understandable Response: Thank you very much for your suggestion. We have rephrased in the article. Please see line 21 in the revised manuscript. 2. Lines 27 MB you have to introduce the whole name before using abbreviations Response: Thank you very much for your suggestion. We have introduced after the “MB”. 3. Line 45-48 please rephrase, the sentence is too long. Response: Thank you very much for your suggestion. We have rephrased in the article. Please see lines 44-46 in the revised manuscript. 4. Line 48-51 is the repetition of the previous sentence Response: Thank you very much for your suggestion. We have deleted this repeated sentence. 5. Line 51-53 improve the English Response: Thank you very much for your suggestion. We have improved in the revised manuscript. 6. Line 57 please change direction to with towards Response: Thank you very much for your suggestion. We have changed ‘direction to’ with ‘towards’. Please see line 53 in the revised manuscript. 7. Line 59 please delete to after attached Response: Thank you very much for your suggestion. We have deleted ‘to’ after attached. Please see line 55 in the revised manuscript. 8. Line 60-61 should be rephrased, For example Suzuki et al used polyethylene... Response: Thank you for your suggestion. We have made corrections in the revised manuscript. 9. Line 62 pH sensitive indicator labels are pH sensors, right? Response: Thank you for your question. And we have changed in the article. 10. Line 76-80 please rephrase in more English manner Response: Thank you very much for your suggestion. We have improved in the revised manuscript. 11. Line 81-83 the goal or aim of the research should be stressed and point out novelty, not just a single statement. Response: Thank you very much for your suggestion. We have improved in the revised manuscript. Please see lines 77-81 in the revised manuscript. 12. Line 91-93 please modify and explain better. Response: Thank you very much for your suggestion. We have made some improvement in the revised manuscript. 13. Line 96-102 should be deleted from here because this is explanation and not experimental procedure Response: Thank you for your suggestion. We have deleted these sentences in the revised manuscript. 14. Line 103-105 please rewrite, it is ethanolic or alcohol solution of MR was prepared by dissolving 1g of MR in 1 L of how much was the ethanol percentage and so on... Response: Thank you for your suggestion. We have provided further explanation in the article. 15. Line 106-109 also explain better, you can not say certain amount you should give the exact amount, then it was heated to 70 degrees or just the bath was at 70 degrees, then how long you kept it and so on. Response: Thank you for your suggestion. We have provided further explanation in the article. 16. Line 112 culture dish do you mean Petri dish? Response: Thank you for your question. We have made corrections. 17. Line 113 intelligent label indicator film can you use one uniform name? there is no need to put extra words to sound fancy. Response: Thank you for your suggestion. We have taken abbreviations for each label. 18. Line 118 anthocyanidin should be with small letter Response: Thank you for your question. We have made corrections. But this part was deleted according to the next suggestion (19). 19. Line 118-119 it is not procedure, please delete Response: Thank you for your suggestion. We have deleted these sentences in the revised manuscript. 20. Line 120 mulberry extract, please explain did you buy this extract or you prepared it yourself? If you did prepare it explain how, is it in liquid or powder state? Response: Thank you for your suggestion. We have provided further explanation in the article. 21. Line 121-123 please explain how did you prepare the PVA and starch solutions. Response: Thank you for your suggestion. We have provided further explanation in the article. 22. Line 123 please delete rich in anthocyanin and be consistent whether you will use glycerin or glycerol Response: Thank you for your suggestion. We have made improvement in the article. 23. Line 124 is the total weight of the mixed solution or the biopolymers weight because you are supposed to add plasticizers in respect to the polymer and not the weight/volume of the solution. Response: Thank you for your suggestion. We have made corrections. 24. Line 124 please change defoaming with defoamed Response: Thank you for your suggestion. We have made corrections. 25. Line 125 please specify is the glass mold a Petri dish or other type of mold and is the glass non-sticky and can you peel of the film easily? Response: Thank you for your suggestion. We have made corrections. And it is a Petri dish in fact. 26. Line 127-128 when you mention that it was prepared like previous method, is this about the part of 1cm2 squares that you cut out? explain better Response: Thank you for your suggestion. We have provided further explanation in the article. 27. Line 129 please give the exact amount and write 1% acetic acid Response: Thank you very much for your suggestion. We have made corrections. 28. Line 134 what do you mean by baked? did you mean dry/ dried? Response: Thank you for your question. We have made corrections in the revised manuscript. Please see line 125. 29. Line 140 please change shown with explained or written Response: Thank you very much for your suggestion. We have changed ‘shown’ with ‘explained’. 30. Line 140-142 please delete this sentence, it is result or statement and not procedure Response: Thank you for your suggestion. We have deleted these sentences in the revised manuscript. 31. Line 147 it should be the extract was properly diluted... Response: Thank you for your question. We have made corrections in the revised manuscript. 32. Line 162 how did you record the color change? did you use colorimeter, what type? Response: Thank you for your question. We have provided further explanation in the revised manuscript. 33. Line 165 %(v/v) of what? Response: Thank you for your question. The original meaning of this sentence is that the results are presented in percentage form. We have made corrections in the revised manuscript. 34. Line 167-169 please delete, not procedure Response: Thank you for your suggestion. We have deleted these sentences in the revised manuscript. 35. Results in 3.2 ;3.3 and 3.4 need comparison with other research Response: Thank you for your suggestion. We have provided further explanation and improvement in the revised manuscript. 36. Line 366-368 please rewrite Response: Thank you very much for your suggestion. We have rewritten this part. 37. Line 375-381 please rewrite and please dont start consecutive sentencec in the same way. Response: Thank you very much for your suggestion. We have rewritten this part.

Reviewer 3 Report

After carefully reading the manuscript the following minor suggestions for improvement were formulated:

line 41- "The freshness of fruits and vegetables has significantly affected.." 

As this is a continous action I suggest changing the tense into "...significantly affects.."

line 44 "food in storage" --> "stored food"

line 52 "whether there is rotten" to be replaced with somtehing similar to "inandequate microbiological quality"

Line 52 "This label" may be missunderstood, readers could interpret it as <food packaging label>

Line 57 instead of "people" "researchers" maybe be more appropriate

Line 60: "For example, use polyethylene active" I thing a "the" is missing: "For example, the use of polyethylene active"

Line 76: " Some scholars have studied the freshness indicator labels of fresh green bell peppers (Chen et al. 2018)." I suggest moving it after the following phrase, in line 83. Also, some conclusions for the previous studies should be added, e.g. increase of freshness duration, longer shelf life periods.

Line 92- "After packaging, stored at 4 ℃,.." it seems that the subject is missing. I suggest ""After packaging, samples were stored at 4 ℃,..". Also, adding information on RH of the storage environment would be relevant for the study. "and measured the relevant indexes"--> "the relevant indexes were measures..."

Line 105-110: "A certain amount of distilled water" The final concentration of of glycerol and methylcellulose is enough. "put it in a 70 ℃ water bath" Please indiacte the duration of the thermal treatment. A more scientific description of the steps would be welcomed here. "certain amount of indicator" Please indicate the precise quantity. "high speed" -Please indicate the exact value in rpm.

Lines 112 & 125: more information on the parameters/ type of US (e.g bath/ sonication probe) are necessary for "After ultrasonic defoaming,"

Lines 112 &  125: "culture dish" and "glass mold" means "Petri dish"?

Line 113 "dried at 45℃ for 8 hours" needs more information (e.g. convective drying or some other method)

Line 129: "A certain amount of chitosan" is not scientifficantly appropriate

Line 134: maybe instead of "baked" it's better to use "dried"

Line 134-136: "and baked in an electric constant temperature blast drying oven at 45 ℃  for 15 hours. Then removed the dry film to get the final composite film. Named this indicator film K-H." All these 3 prases could be merged into one, shorter

Line 166: "2.6 Quality test of vegetable samples 

A series of quality changes will occur during the storage of fruit and vegetable products. In addition to color change, nutrient content change and sensory evaluation are also very important (Chitrakar et al. 2019; Zhang et al. 2019)."  The information presented under 2.6 should refer to methodology used, plase update the information

Line 209: "before to understand"--> "before understanding.."

Lines 225, 233 and elsewhere in the text: No need to use preposition "the" (The Fig. 1), "Figure 1" is good enough. The tense should be present "Fig 1 indicates..."

Line 228: "Other study had shown similar result (Kan et al. 2022)." Please indicate exactly the similar results and compare them by discussing the results obtained in your research

Line 229: Only one determination was performed or at least a duplicate for the value 173.45 mg/g? If the latter applies please indicate the STDEV values.

Line 241: "low temperatures" mean "refrigeration temperatures"?

Line 244-253: "resulting in the CO2 concentration in the packaging gradually increased with the extension of storage time" Indicate the % increase or the difference between values, compare them with other reported values. For color difference the same comment.

Line 251: "insignificant change in CO2 concentration" Please comment the results in light of the statistical significance.

Line 286: "It can be seen from the Table 5 that the color of the label changed  with the change of vegetables and was related" Please rephrase so that is makes sense, now it seems something is missing.

Line 301: "stale samples" I don't think "stale" is the right term for vegetable decaying. 

Fig 4 VC content of vegetables during storage has a scale up to 150, even though the higher values are below 100.

Suggestions for grammar and meaning improvement of English were made.

Author Response

After carefully reading the manuscript the following minor suggestions for improvement were formulated:

  1. line 41- "The freshness of fruits and vegetables has significantly affected.."

As this is a continous action I suggest changing the tense into "...significantly affects.."

Response: Thank you for your suggestion. We have made corrections to this part.                                                       

  1. line 44 "food in storage" --> "stored food"

Response: Thank you for your suggestion. We have made corrections to this part.

  1. line 52 "whether there is rotten" to be replaced with somtehing similar to "inandequate microbiological quality"

Response: Thank you for your suggestion. We have made corrections to this part.

  1. Line 52 "This label" may be missunderstood, readers could interpret it as <food packaging label>

Response: Thank you for your suggestion. We have made corrections to this part.

  1. Line 57 instead of "people" "researchers" maybe be more appropriate

Response: Thank you for your suggestion. We have made corrections.

  1. Line 60: "For example, use polyethylene active" I think a "the" is missing: "For example, the use of polyethylene active"

Response: Thank you for your suggestion. We have made corrections.

  1. Line 76: " Some scholars have studied the freshness indicator labels of fresh green bell peppers (Chen et al. 2018)." I suggest moving it after the following phrase, in line 83. Also, some conclusions for the previous studies should be added, e.g. increase of freshness duration, longer shelf life periods.

Response: Thank you for your suggestion. We have made corrections.

  1. Line 92- "After packaging, stored at 4 ℃,.." it seems that the subject is missing. I suggest ""After packaging, samples were stored at 4 ℃,..". Also, adding information on RH of the storage environment would be relevant for the study. "and measured the relevant indexes"--> "the relevant indexes were measures..."

Response: Thank you for your suggestion. We have made corrections.

  1. Line 105-110: "A certain amount of distilled water" The final concentration of of glycerol and methylcellulose is enough. "put it in a 70 ℃ water bath" Please indiacte the duration of the thermal treatment. A more scientific description of the steps would be welcomed here. "certain amount of indicator" Please indicate the precise quantity. "high speed" -Please indicate the exact value in rpm.

Response: Thank you for your suggestion. We have made revisions and additional explanations to this section. The indicator is used to adjust the pH value, and the specific dosage is not clear, but the specific pH value that needs to be adjusted is explained later.

  1. Lines 112 & 125: more information on the parameters/ type of US (e.g bath/ sonication probe) are necessary for "After ultrasonic defoaming,"

Response: Thank you for your suggestion. We have made additional explanations to this section.

  1. Lines 112 & 125: "culture dish" and "glass mold" means "Petri dish"?

Response: Thank you for your question. We have made corrections.

  1. Line 113 "dried at 45℃ for 8 hours" needs more information (e.g. convective drying or some other method)

Response: Thank you very much for your suggestion. We have added more information here.

  1. Line 129: "A certain amount of chitosan" is not scientifficantly appropriate

Response: Thank you for your question. We have made corrections.

  1. Line 134: maybe instead of "baked" it's better to use "dried"

Response: Thank you for your question. We have made corrections.

  1. Line 134-136: "and baked in an electric constant temperature blast drying oven at 45 ℃ for 15 hours. Then removed the dry film to get the final composite film. Named this indicator film K-H." All these 3 prases could be merged into one, shorter

Response: Thank you for your question. We have made appropriate modifications and deletions to this section.

  1. Line 166: "2.6 Quality test of vegetable samples

A series of quality changes will occur during the storage of fruit and vegetable products. In addition to color change, nutrient content change and sensory evaluation are also very important (Chitrakar et al. 2019; Zhang et al. 2019)." The information presented under 2.6 should refer to methodology used, plase update the information

Response: Thank you for your suggestion. We have provided the reference.

  1. Line 209: "before to understand"--> "before understanding.."

Response: Thank you very much for your suggestion. We have made corrections.

  1. Lines 225, 233 and elsewhere in the text: No need to use preposition "the" (The Fig. 1), "Figure 1" is good enough. The tense should be present "Fig 1 indicates..."

Response: Thank you very much for your suggestion. We have made corrections.

  1. Line 228: "Other study had shown similar result (Kan et al. 2022)." Please indicate exactly the similar results and compare them by discussing the results obtained in your research

Response: Thank you for your suggestion. We have added some information.

  1. Line 229: Only one determination was performed or at least a duplicate for the value 173.45 mg/g? If the latter applies please indicate the STDEV values.

Response: Thank you for your suggestion. We have indicated the STDEV values

  1. Line 241: "low temperatures" mean "refrigeration temperatures"?

Response: Thank you for your question. It means "refrigeration temperatures", and we have made correction.

  1. Line 244-253: "resulting in the CO2 concentration in the packaging gradually increased with the extension of storage time" Indicate the % increase or the difference between values, compare them with other reported values. For color difference the same comment.

Response: Thank you for your suggestion. We have added some information.

  1. Line 251: "insignificant change in CO2 concentration" Please comment the results in light of the statistical significance.

Response: Thank you for your suggestion. We have added some information.

  1. Line 286: "It can be seen from the Table 5 that the color of the label changed with the change of vegetables and was related" Please rephrase so that is makes sense, now it seems something is missing.

Response: Thank you for your suggestion. We have rephrased this section.

  1. Line 301: "stale samples" I don't think "stale" is the right term for vegetable decaying.

Response: Thank you very much for your suggestion. We have replaced "stale samples" with "rotten samples".

  1. Fig 4 VC content of vegetables during storage has a scale up to 150, even though the higher values are below 100.

Response: Thank you for your question. For the convenience of observation and aesthetic considerations, the highest value of the y-axis in the figure is set to 150, but it does not affect the observation.

Reviewer 4 Report

The idea is good and very interesting but I have a few comments: 

1. The abstract was poor and didn't cover all manuscript parts, so, it would be nice if you rewrite it to contain all the tests you have done and their results.

2. The abstract should not contain abbreviations or you should explain it first such as, (MB).

3. Lines 27-28: It's unclear and needs to be rewritten.

4. The methods of label preparation were confusing (you give me the impression it is a top secret !). you should write the methods as you did already to give others a chance to repeat this work and develop it.

5. Also, you wrote that you made an ultrasonication in preparing labels but you didn't mention the conditions, why?

6. The title "2.2.3. preparation and use of the intelligent label" would be good if you changed to the usage of the intelligent label because the previous title is about the preparation.

7. L, a, and b should be written in italic.

8. It would be nice if you merged the color of the label and fruit determinations under one title in separate two paragraphs.

9. The results also were written in a poor way and missing the discussions.

10. Th and rd of the day should be superscript.

11. I wonder why you didn't analyze the sensory and delta E results statistically, it's very important to do it.

12. Finally, it would be better if you added photos of pepper fruit types during storage.  

Minor editing of English language required

Author Response

The idea is good and very interesting but I have a few comments:

  1. The abstract was poor and didn't cover all manuscript parts, so, it would be nice if you rewrite it to contain all the tests you have done and their results.

Response: Thank you very much for your suggestion. We have rewritten the abstract.

  1. The abstract should not contain abbreviations or you should explain it first such as, (MB).

Response: Thank you for your suggestion. Due to the complexity of label names, abbreviations are used, but detailed explanations are provided after MB.

  1. Lines 27-28: It's unclear and needs to be rewritten.

Response: Thank you very much for your suggestion. We have rewritten this part.

  1. The methods of label preparation were confusing (you give me the impression it is a top secret !). you should write the methods as you did already to give others a chance to repeat this work and develop it.

Response: Thank you very much for your suggestion. We have added some explanations.

  1. Also, you wrote that you made an ultrasonication in preparing labels but you didn't mention the conditions, why?

Response: Thank you for your question. Because ultrasound is used for defoaming, there are no very clear requirements.

  1. The title "2.2.3. preparation and use of the intelligent label" would be good if you changed to the usage of the intelligent label because the previous title is about the preparation.

Response: Thank you very much for your suggestion. We have changed the title.

  1. L, a, and b should be written in italic.

Response: Thank you very much for your suggestion. We have corrected this part.

  1. It would be nice if you merged the color of the label and fruit determinations under one title in separate two paragraphs.

Response: Thank you very much for your suggestion. We have merged them under one title.

  1. The results also were written in a poor way and missing the discussions.

Response: Thank you very much for your suggestion. We have rewritten this part.

  1. Th and rd of the day should be superscript.

Response: Thank you for your suggestion. We have corrected this part.

  1. I wonder why you didn't analyze the sensory and delta E results statistically, it's very important to do it.

Response: Thank you for your suggestion. We believe that the change in green color of vegetables was particularly important throughout the entire storage process, so using values of L*, a* and b* values were more intuitive.

  1. Finally, it would be better if you added photos of pepper fruit types during storage.  

Response: Thank you for your suggestion. We have added photos after packaging, which were more meaningful than just photos of pepper.

Round 2

Reviewer 1 Report

Dear Author,

The suggestion I made below was not taken into account in the first round evaluation of the article.

No detailed explanation was made regarding the statistical analysis.

L 217: The statistical analysis part of the study is very weak. Authors should provide more detailed explanations for the reader. In the study, the first Normal distribution and homogeneous tests are applied to the data for statistical analysis. According to the result obtained from this, parametric or non-parametric analysis is performed. However, these were not mentioned in this study. According to which criteria statistical analyzes were made. Authors should explain in detail.

Adequate

Author Response

Reviewer #1:

Dear Author,

The suggestion I made below was not taken into account in the first round evaluation of the article.

No detailed explanation was made regarding the statistical analysis.

L 217: The statistical analysis part of the study is very weak. Authors should provide more detailed explanations for the reader. In the study, the first Normal distribution and homogeneous tests are applied to the data for statistical analysis. According to the result obtained from this, parametric or non-parametric analysis is performed. However, these were not mentioned in this study. According to which criteria statistical analyzes were made. Authors should explain in detail.

Response: Thank you for your suggestion. We have provided further explanation and improvement in the revised manuscript.

Statistix 9.0 was used for one-way ANOVA and Tukey multiple comparison test was used for significance analysis. The different letters in the figure and table indicated significant differences between samples (p<0.05), and the data was represented by mean±standard deviation. GraphPad Prism 8.0 was used for drawing. Origin 2021 was used for principal component analysis. SPSS 16.0 was used for fisher discriminant analysis..Number of training (modeling) samples: The number of predicted samples was 2:1. When modeling, chose to leave one method as the cross validation method.

Please see lines 221-232 in the revised manuscript.

Reviewer 4 Report

All the modifications a I asked were done 

Minor editing of English language required

Author Response

Reviewer #4:

All the modifications a I asked were done.

Minor editing of English language required.

Response: Thank you very much for your suggestion. We have further improved the English language of the article. Revised portion are marked in red in the revised manuscript.